# Development of an Electrically Conductive MDF Panel—Evaluation of Carbon Content and Resin Type

**DOI:** 10.3390/polym15040912

**Published:** 2023-02-11

**Authors:** Christof Tschannen, Ali Shalbafan, Heiko Thoemen

**Affiliations:** 1Institute for Materials and Wood Technology, Bern University of Applied Sciences (BFH), CH-2500 Biel, Switzerland; 2Department of Wood and Paper Science and Technology, Faculty of Natural Resources and Marine Sciences, Tarbiat Modares University, Noor P.O. Box 46414-356, Iran

**Keywords:** wood-based panel, medium density fibreboard, carbon fibres, electrical conductivity, furniture design, urea formaldehyde, pMDI (polymeric methylene diphenyl diisocyanate)

## Abstract

Electronics in furniture and construction materials, in particular technologies which allow for a flexible and cable-free connection of electronics in such materials, are gaining broader interest. This study shows a further development of a concept to obtain highly conductive medium-density fibreboard panels (MDF) for furniture application. MDF were produced using two mixing processes (wet and dry) for wood and carbon fibres to investigate the effects of resin type (urea formaldehyde (UF) and polymeric methylene diphenyl diisocyanate (pMDI)) and carbon fibre content on their mechanical, physical, and electrical properties. Overall, wet mixed fibres showed better electrical but reduced mechanical properties. Modulus of elasticity (MOE) and bending strength (MOR) values of 3500 MPa and 35 MPa, respectively, and internal bond (IB) values of 0.45 to 0.65 MPa with electrical conductivities of up to 230 S/m were achieved. The technology has been successfully implemented in a demonstration object showing the application in a small piece of furniture.

## 1. Introduction

The use of electronics in furniture is on a rising trend and has reached unprecedented heights with more and more applications to follow, specifically in the field of smart applications [1,2,3,4,5,6,7,8,9]. Powering electronics such as light-emitting diodes (LED) for interior lighting can be achieved primarily with two methods: either power is provided over a wire or rail or it is provided by a battery directly at the place of the consumer [5,8]. Both variations have their advantages and disadvantages. Either one of these variations requires a fixed, installed light source and usually the positioning of said light cannot be relocated after the initial construction of the furniture. The standard application today is to use LED-strips to illuminate the entire piece of furniture, and thus the arrangement of the furniture is more flexible but still locked to the initial positioning of the strips. A new concept is to use the structure of the furniture itself as a conductor, i.e., the wood composite. The concept was first shown in 2019 [10,11]. The idea behind the concept was to provide the electricity by conducting a current through the surface layers of a three-layered particleboard. With this technology, the concept of the cable-free conduction of electricity in furniture was established. Part of the concept was the very high flexibility in positioning of interior lighting since the entire furniture panel is conducting electricity. A light source in the form of an LED can be positioned in whatever arrangement the user wishes and does not require any further cable management within the furniture. The technology also allows for using more than one LED in a panel.

Naturally, wood and wood-based products have rarely any electrical conductivity and this is mostly due to their being affected by moisture content [12]. There are some methods known to improve the electrical properties of wood and wood-based products. A non-complete list of state-of-the-art methods shall be mentioned here. The effect of wood preservative, often containing salts originating from copper for impregnation is known to affect the electrical properties of wood [13]. An impregnation process was also used by Trey et al. [14] to impregnate the lumen of wood with an intrinsically conductive polymer. In their research, Agarwal et al. [15,16] showed the coating of wood-based cellulose fibres with conductive polymers and carbon nano tubes to create conductive paper which could be scaled to some extent to wood-based products. Another publication [17] showed the coating of wood with a thin conductive copper nanowire to conduct electricity on the surface of wood. Moreover, it is possible to carbonize biomass to generate conductive fibres to produce conductive composites [18,19]. Furthermore, it is possible to create a conductive surface of wood by graphitization of the surface using laser technology [9]. Lastly, different studies showed the carbonization of a complete wood-based fibreboard [20,21,22]. A number of publications [15,23,24,25,26] shows the use of either carbon fibre (CF) or carbon nanotubes as conductive filler in wood-based panels or paper to improve selected characteristics of wood-based panels such as mechanical properties, electro-magnetic shielding, and thermal and electric conductivity.

To create a conductive furniture panel, a conductive layer is needed [10]. This layer can be a surface coating or may consist of the panel structure itself. A conductive layer as a separately added layer is shown by Łukawski et al. [24] as an example. The approach in this study is to create a conductive layer from a homogeneous mixture of mainly wood fibres (WF) and carbon fibres (CF). Metal fibres as a conductive additive are mostly unusable as they have a natural tendency to corrode, even more so in the environment of a wood-based panel where there is moisture, heat, and low pH-values present during processing: all beneficial towards corroding metal [27]. Certain sorts of metal fibres do not suffer from these conditions as much but have other disadvantages such as high price (i.e., precious metals) and hardness (i.e., stainless steel), which leads to increased wear of tools when processing the panels with wood machinery [28]. Carbon fibres do not suffer from these disadvantages. Carbon fibres have high electrical conductivity [29] of up to 9.09 × 10^5^ S/m and are relatively cheap compared to other conductive materials. CF are known to have an inert surface which do not allow for any interaction with most adhesives [30], which includes UF, which is mainly used in wood-based panel manufacturing [31]. Any conductive additive must be processable with the standard processes used in the wood-based panel production, and once integrated into the panel, it should allow to form a conductive network as it is described in the field of conductive polymers [32,33]. In a previous study by Tschannen et al. [10], it was demonstrated that it is challenging to obtain a homogeneous mixture of WF and CF. In said study, the mixture showed a very inhomogeneous structure which was made responsible for the weak results, specifically the mechanical and electrical properties. To obtain a better mixture of WF and CF, a new method was needed. In their work, Pourjafar et al. [26] showed that a homogenous mixture of WF and CF is possible when processed in water. First the CF are dispersed in water and afterwards the WF is added under continuous stirring.

In the present study, we want to investigate the effect of the resin type, mixing method, and carbon fibre content in relation to mechanical, physical, and electrical properties. Therefore, the resin system is varied, using a UF and a pMDI resin. Two processes to mix the wood and carbon fibres are employed: a dry mixing and a wet mixing process. The effects of these variables on the electrical properties of the wood-based panels are investigated in this study.

## 2. Materials and Methods

### 2.1. Materials

#### 2.1.1. Wood Fibres

Wood fibres (WF) of laboratory grade quality were obtained from the Institut für Holztechnologie Dresden gemeinnützige GmbH (IHD Dresden), Dresden, Germany, and produced with their laboratory refiner. The fibres consist of coniferous trees, mainly spruce (*Picea abies* (L.) Karst.), and were stored in dry conditions before using them. The fibres had an approximate moisture content of 6%.

#### 2.1.2. Resin

Urea formaldehyde (UF) resin with the trade name Kaurit Leim 345 with an average solid content of 67% was used for the main part of the work presented in this paper. pMDI with the trade name Lupranat M 20 s was used for some parts of the experiments. Both resin systems were kindly provided by BASF SE, Ludwigshafen, Germany.

#### 2.1.3. Carbon Fibres

Chopped carbon fibres (CF) were obtained from Suter-Kunststoffe AG, Fraubrunnen, Switzerland, with the trade name SCS Carbon-Schnitzel 3 mm and with nominal length of 3 mm and carbon content of >95%. The fibres consist of a mixture of high-tensile fibres (>90%) and high-modulus fibres (<10%) according to the material data sheet by Sutter-Kunststoffe AG, Fraubrunnen, Switzerland. The CF are in a loose state but with a tendency of sticking to each other.

### 2.2. Methods

#### 2.2.1. Mixing of Wood Fibres with Carbon Fibres (Wet Process Mixing)

Chopped CF with the nominal length of 3 mm were mixed with WF in a water suspension (Figure 1). This process was followed to achieve a homogeneous mixture of wood and carbon fibres. In one batch, 300 g of solid material (WF and CF) were mixed in approximately 10 kg of water, thus the fibres-to-water ratio was 3%. There was no further optimization and assessment of the fibres-to-water ratio in this study. A stirring vessel was first filled with tap water. The needed CF content was added to the water and mixed with a stirrer (cement stirring rod) at approximately 1000 rpm for 5 min. After this step, the WF were added to the suspension of CF and mixed for another 10 min at 1000 rpm until a homogeneous mixture was obtained. Then the mixed wood/carbon fibres were drained from the water and dried in a recirculation air-dryer at 60 °C for 48 h. This step was repeated several times since the process allowed for a maximum batch size of 1 kg dry wood/carbon fibre mix with the dryer as a bottleneck. Mixtures containing 1, 3, 5 and 10 wt% of CF based on the dry wood mass were prepared using this method. For both methods (wet and dry mixing) the CF addition was based on the oven dry mass of the WF. The CF substituted the WF based on their mass, i.e., 5% wt% CF means that 5 wt% WF was substituted by CF. This calculation was done before the actual mixing process (wet and dry).

##### Mixing of Wood Fibres with Carbon Fibres (Dry Process Mixing)

Mixtures of wood and carbon fibres containing 5 and 10 wt% of CF based on the dry wood mass were prepared using a dry mixing process. A specially developed laboratory machine was used for this process; the machine is basically a container where the WF and CF are mixed using compressed air.

#### 2.2.2. Manufacturing of MDF panels

Thin MDF panels (A–J) with a thickness of 5 mm and target density of 750 kg/m^3^ were manufactured on a laboratory scale using the previously prepared mixtures of wood/carbon fibre and both available resin types (UF and pMDI); see Table 1. The process included two steps: (1) addition of UF or pMDI to the previously prepared mix of wood/carbon fibres and (2) pressing on the final panel using a hot press. In addition, panels with a 5-layered structure (K) were manufactured using the same mixture of WF and CF and UF resin.

The resin content for the panels was 15 wt% (UF) and 8 wt% (pMDI) based on the dry fibre mass. For the panels manufactured with UF resin, ammonium sulphate in an aqueous solution (40%_*w*/*w*_) was added at a rate of 1 wt% based on the solid content of the added resin. The materials were mixed using a rotary mixer (Lödige FM 300D, Gebrüder Lödige Maschinenbau GmbH, Paderborn, Germany). For the hot pressing, a laboratory hot press (Höfer HLOP 210, Höfer Presstechnik GmbH, Taiskirchen, Austria) was used at a press plate temperature of 180 °C and operating with a press factor of 15 s/mm and distance control, resulting in a press time of 75 s and a maximum pressure (average) of ca. 4.4 MPa for all boards.

In addition to the thin MDF panels, conductive 5-layered MDF panels with a thickness of 16 mm were also manufactured using UF-resin to produce a small conductive piece of furniture as demonstration object. For those 5-layered MDF panels, the layers 2 and 4 had approximately 1.5 mm thickness and contained the mixture of wood and 5 wt% carbon fibres (wet mixed process). The remaining layers, 1, 3 and 5, were made from wood fibres without any addition of carbon fibres. The thickness of layers 1 and 5 was set to 2 mm each.

#### 2.2.3. Electrical Properties of MDF panels

The electrical properties of conductive MDF panels were tested on panels A-J as in Table 1. The MDF panels were acclimatized prior to the testing in the standard climate of 20 °C and 65% relative humidity. The testing did not follow a standard but is based on the so called “four-probe method” [34]. The method is shown as a schematic in Figure 2. A thin conductive copper tape (3M 1181) is joined on two opposite edges of the sample that it covers, at the complete edge and some small area on the surface and bottom of the panel. The copper tape was joined by applying an even force alongside the tape. Then, a copper wire is soldered onto the copper tape approximately in the centre of the tape on both edges connecting one (-) to the DC power supply and the other to the first multimeter (Agilent U1241B) measuring the current (I). The second cable from the multimeter measuring the current (+) also connects to the DC power supply. For the power supply, a current of approximately 1 Ampere is set, and the voltage is set to a value that the resulting power equals ca. 5 Watt. Depending on the resistivity of the panel, the voltages are higher or lower.

Then a second multimeter (same brand and type) is used to measure the voltage (U). For this task, two electrodes are connected to the multimeter. The electrodes are two thin copper plates, ca. 30 mm × 30 mm × 0.5 mm. The copper electrodes are then held onto the surface of the panel with a defined distance in between them. The first measurement starts 30 mm apart with the centre of the board as central-reference points. Ten measurements are taken in total, with each measurement having 10 mm more distance to the centre of the panel for each electrode (increments of 20 mm between electrodes). For each measurement, the current (which was nearly constant) and voltages were measured and recorded. Using Equation (1), the resistance can be calculated at each position where a measurement was taken.
(1)U=R×I

The information about the resistance (*R*) can then be further processed to calculate the resistivity (ρ) by using Equation (2), where *w* is the width (perpendicular to the measuring path), and *t* is the thickness of the panel. The distance in between the electrodes is *l*.
(2)$$\rho =\left(R\times w\times t\right)/l$$

For each measurement point, the resistivity should theoretically be constant, which serves as a validation method for the homogeneity of the panel’s electrical properties. The results for the electrical properties are then shown as resistivity resulting as an average of 10 measurements. 

Some data in this paper are shown as conductivity (σ) in S/m. The conductivity is the reciprocal value of the resistivity and can be calculated by using Equation (3).
(3)σ=1/ρ

#### 2.2.4. Mechanical and Physical Properties of MDF panels

Based on the electrical properties obtained, mechanical and physical properties of selective MDF panels, containing 3 and 5 wt% of CF resinated with UF resin, were measured. Internal bond strength (IB) and density was conducted following the description in SN EN 319:1993 [35] and SN EN 325:2012 [36] using a universal testing machine (ZwickRoell Z030, ZwickRoell, Ulm, Germany). For each panel type, 9 samples were tested.

Modulus of elasticity (MOE) and modulus of rupture (MOR) were also tested by following the standard SN EN 310:1993 [37] and using the same test machine as for the IB testing. In total, 10 samples per panel type were tested.

Furthermore, the thickness swelling of the panels was also tested following the description of SN EN 317:1993 [38]. For each panel type, 10 samples were tested.

Lastly, the density and density profile of the panels were tested using a density profile scanner (DAX 5000, Fagus-GreCon Greten GmbH & Co. KG, Alfeld-Hannover, Germany). From each panel, 9 samples were measured. The samples were cut to 50 mm × 50 mm for this purpose.

#### 2.2.5. Microscopy (Scanning Electron Microscopy)

Scanning electron microscope (SEM) was used to investigate and qualitatively evaluate the distribution of CF within the conductive MDF panel in the core layer of samples previously tested for IB. A Hitachi tabletop microscope TM3030 with 15 kV was used for this assessment. The sample surface was previously sputter coated with a thin gold layer of approximately 20 nm.

#### 2.2.6. Optical Assessment of Wood/Carbon Fibre Mixture

The quality of the mixture of wood and carbon fibre was furthermore carried out by an optical assessment of the panel’s core layer after testing for IB strength. The samples usually break in the core of the panel and reveal the material distribution. The interest here is whether the materials (WF and CF) are homogenously mixed or if there are agglomerates of CF visible within the mixture.

#### 2.2.7. Furniture Manufacturing

To prove applicability of the concept, a small piece of furniture consisting of two cubes was manufactured using the 5-layered conductive MDF panels. The surface of the panels was covered with a wood veneer. The 5-layered structure allowed for processing as is possible with a common MDF. The illumination inside the cubes was managed using LED pins as previously developed for the concept of a conductive furniture panel [10,11].

## 3. Results and Discussion

### 3.1. Electrical Properties

Panels manufactured with UF resin show much lower resistivity than panels manufactured with pMDI using the wet mixing process; see Figure 3a. Despite some extensive research no study was found to confirm this behaviour of lower conductivity using pMDI resin compared to UF. One study mentions that isocyanate hardly conducts any electricity and thus does not influence the electrical properties of wood-based panels [23]. From the application of powder coating where the electrical conductivity of the substrate plays an important role, it is reported that panels using pMDI achieve higher conductivity values than panels bonded with UF or MUF [39]. This however is a contradiction to the results obtained in this study.

The secondary Y-axis in Figure 3a shows the increment of resistivity of panels with pMDI based on the values of panels with UF. The difference between UF and pMDI becomes smaller (ca. 20%) at 10 wt% CF concentration and is significantly increased (ca. 400%) at 1 wt% concentration of CF. However, both panel types have the same behaviour, where panels with increased CF content show lower resistivity values, which is confirmed also by other publications [10,18,25]. Based on these results it was decided to not carry out further experiments using pMDI and instead only using UF resin for further experiments. Furthermore, results for 5 and 10 wt% CF show the most promising results regarding the required electrical properties. Thus, for later tests, only those two concentrations were tested.

Figure 3b shows the comparison between the wet mixing and dry mixing process and the influence on the resistivity of the MDF panels. The data for the wet mixing is the same as shown in Figure 3a. For the dry mixing process new panels were manufactured using the UF resin with the dry mixed fibres at the concentration of 5 and 10 wt% of CF. The concentrations of 1 and 3 wt% of CF were excluded in the test for the dry mixing since the resistivity was considered as not good enough for implementation in a larger scale panel. The mixing process seems to influence the electrical properties of the conductive MDF panels more than the resin did in the first comparison (Figure 3a). The resistivity for both concentration of dry mixed fibres is significantly higher than for the wet mixed fibres with an increase in resistivity of more than 100% at 10 wt% and more than 200% at 5 wt% CF concentration. The increment of resistivity is shown as percentage of the dry mixing process based on the values of the wet mixing process on the secondary Y-axis in the graph.

A clear indication of the bad performance of dry mixed fibres is the comparison between the value for 3 wt% CF using the wet mixing process compared to the 5 wt% CF using the dry mixing process. The resistivity value was lower in panels using 3 wt% wet admixtures of CF + WF (about 0.03 Ωm) compared to that of 5 wt% dry admixture which was about 0.04 Ωm. This is clearly due to the better distribution of CF among WF using a wet mixing process. The orientation and distribution of carbon fibers largely influences the electrical resistivity and conductivity of the composites [25,26]. At a constant electrical resistivity value, better distribution of CF among WF can lead to considerably less CF content in the composites. This is highly desirable from economic and environmental perspectives.

The laboratory-made conductive MDF panels all showed some levels of conductivity ranging from 0.31 S/m up to 229.2 S/m, as summarized in Table 2. In direct comparison to the conductivity of CF, the produced MDF panels have a much smaller conductivity. The conductivity of CF is in a range of 5.56 × 10^4^ S/m to 9.09 × 10^5^ S/m according to the literature [29]. However, the obtained values are much better than shown in previous studies where particleboards instead of MDF panels were made [10].

According to these findings, the improvement in conductivity is related to the improved mixing process (wet mixing) and change from wood particles to WF as the raw material. The use of WF instead of wood particles, even in a dry mixing process, has already improved the electrical properties. In the previous study [10], the resistivity for a particle board containing 10 wt% of CF resulted in 0.016 Ωm, and the results of the dry mixed WF with 10 wt% CF from this study resulted in 0.009 Ωm. It is assumed that due to the similar shape of WF and CF, a better entanglement is possible than with wood particles and CF. Looking at the differences resulting from wet and dry mixing processes presented in this paper while also considering the results from the previous study, it can be concluded that both factors contribute towards these favourable results.

### 3.2. Mechanical and Physical Properties of MDF panels

The mean density of different conductive MDF panels and their density profile (DP) is shown in Figure 4. A clear trend is visible towards the conclusion that higher concentrations of CF lead to an overall reduction in density, where the samples with 10 wt% have the lowest density and those with 3 wt% have the highest. This is plausible since the CF addition is achieved by substitution of WF by CF in weight percent. Consequently, samples with higher CF content have less wood overall. Thus, when the samples are stabilized in standard climate conditions, where water is absorbed by the samples due to hygroscopic behaviour of wood, those with a higher CF concentration can effectively absorb less water, resulting in a lower mean density. The density profile itself is the same for all panel types, just with a different maximum and minimum. The density peak at the panel surface is not as pronounced as it could be. This is well known from the equipment used, since additional aluminium plates with a thickness of 6 mm are used on top and bottom of the panel during pressing, delaying heat transfer from the press plates into the panel. However, optimization for a well-defined density profile was not the aim of these experiments and not further analysed.

The characterized mechanical and physical properties of the conductive panels are shown in Figure 5. The thickness swelling (a) is for both dry and wet mixed panels for 24 h at a good level, below the required 30% according to the standard EN 622-5:2009 [40] (indicated as black bar). The explanation for the slightly lower values of the wet mixed fibres is due to the washing out of fine fibres during the wet mixing process. The fine fibres have a higher specific surface and are more accessible by water molecules, hence the higher swelling of the dry mixed fibres where the fine fibres are still present. Similar observations were made in a different study [41] where smaller particles resulted in higher thickness swelling. 

The internal bond properties requirements are also set in EN 622-5:2009 at 0.65 N/mm^2^ for a board of 5 mm thickness Figure 5b. Only the dry mixed fibres at 5 wt% CF with average value of 0.67 N/mm^2^ can achieve the required strength. The remaining samples are below the required strength values. For the dry mixed samples, a lower strength value was expected due to the agglomerates of carbon fibres which are clearly visible by optical assessment of the mixed fibres. Even though the agglomerates are present and potentially cause areas with higher stress, meaning that it causes weak points, the measured internal bond was still higher than with the wet mixed panels where a very homogenous mixture was observed. For this behaviour, two reasons are possible. First, when using UF-resin for bonding wood and carbon fibres usually no adhesion of the resin can be obtained on the surface of carbon fibres due to the lack of OH groups and their inert behaviour [30]. The bond within the panel is obtained purely between wood fibres. Thus, the carbon fibres are simply mechanically interlocked in between bonded wood fibres. The theory is that when the carbon fibres are present in agglomerates, there are bigger areas where a good bond between wood fibres is provided. Only at some spots the carbon fibres disturb that bonding. Similar results were shown by [42] but for CF reinforced plywood. The presence of CF overall weakened the IB values. For wet mixed fibres the carbon fibres are present everywhere which is good for the electrical properties, but they do create weak spots all over the cross section of the conductive MDF panel. The second reason can also be attributed to the washing out of fine fibres during the wet mixing process. The size of fibres and particles are very important to internal bond values [31]. Most likely the fine fibres play an important role in filling gaps between wood fibres and carbon fibres and thus reinforce the inter-fibre bond. However, they are mostly washed out during the mixing process and can no longer contribute to the needed reinforcement of the bond. It can be expected that when a better performing resin system is used, the IB values would be enhanced. However, it is a balancing act between mechanical properties and electrical properties. In the first phase of the experiments, pMDI resin was used and showed worse performance regarding electrical properties, hence UF resin was used instead. A further improvement on the mechanical properties could be achieved by using a pre-treatment on the carbon fibres as shown in various studies [25,30] to improve the bonding properties of polar UF resin with non-polar CF.

The bending properties are lastly shown in Figure 5c,d. Here the opposite of the internal bond properties can be observed. The standard value is defined by EN 622-5:2009 and indicated by the black bar. The achieved properties surpass the required value for both the modulus of elasticity (MOE) and the modulus of rupture (MOR). However, the content of carbon fibres did not show a significant effect on the properties. For wet mixed fibres, the MOE of 3600 N/mm^2^ and MOR of 37 N/mm^2^ were achieved, while the dry mixed fibres were slightly lower with MOE of 3000 N/mm^2^ and MOR of 30 N/mm^2^. The good distribution of carbon fibres with the wet mixing process has improved the bending properties compared to the internal bond. The same was also reported by Auriga et al. [42], where the IB was weakened but the MOE/MOR was improved on their panels overall. Most importantly, the existence and not the quantity of carbon fibres have improved the bending properties of conductive MDF panels significantly over the standard value, with MOE 2700 N/mm^2^ and MOR 23 N/mm^2^. Similar results were presented by Pourjafar et al. [26], except for the 10 wt% CF loading, where they showed significantly lower values for MOE and MOR. In the present study, higher values were obtained for 10 wt% CF loading with a dry mixing process at around 3000 MPa MOE and 30 MPa MOR. In the study of Hou et al. [25] an improvement of MOE and MOR was shown too; however, their results were overall weaker than the results of the present study at similar CF loadings. They used a similar process (wet mixing) for the material blending and comparable CF but reported the presence of agglomerates at higher concentrations of CF.

In terms of sustainability, the conductive MDF panels must be made from biobased CF. However, biobased carbon fibres often have weaker mechanical properties than their fossil-based counterparts [43] and mostly use cellulose or lignin as precursors. The electrical properties are usually slightly lower than petrol-based fibres but still good enough for the application presented herein. The main challenge for using biobased CF at this stage is their availability.

### 3.3. Microscopy and Optical Assessment of Wood/Carbon Fibre Mixture

For the further assessment of the conductive MDF panels a scanning electron microscope was used: samples from IB testing were further inspected on the revealed surface from the core of the panel. The results for the dry mixed samples are shown in Figure 6. An agglomeration of CF can be seen in (Figure 6a). These agglomerates are visible by eye and appear as black spots. Figure 6b shows the same bundle at the boarders with an increased magnification showing how the CF are interlocked with the WF. This is also further magnified in Figure 6d. Importantly, some areas of the sample do not show any CF (Figure 6c). This confirmed the hypothesis that whenever there are agglomerates, CF are missing in other areas and thus reduce the electrical properties. However, as seen in the results for IB testing, areas where no CF are present significantly improve the IB as a perfect bonding between WF is possible.

The SEM pictures of wet mixed fibres are shown in Figure 7 and confirm the very homogeneous distribution of CF in WF. In the microscopic picture individual fibres can be seen amongst some bundles of 2–10 CF (Figure 7a) as small but loose agglomerates. The individual CF are intertwined with the WF (Figure 7b) and also interlocked between other fibres as demonstrated by the higher magnification in (Figure 7c). The individual CF can be well differentiated from the WF at a high magnification (Figure 7d). However, the CF often hang loosely between the WF and are not densified in the same way as the WF. This is well observed in Figure 7a with the densified WF in the background and some loose CF in the front. Thus, it can be concluded that the CF are not held together by the resin and are solely held back in the panel via interlocking in between the densified WF, hence the poor IB results.

The agglomerates present in the dry mixed samples are clearly visible to the eye. Looking at Figure 8a (wet mixed) and Figure 8b (dry mixed), the difference between the two processes on the example of the fractured surface of IB samples is clearly shown. However, as seen before, the agglomerates do not cause a major weakening to the panels for their mechanical properties.

### 3.4. Conductive Furniture Panel and Application

The conductive MDF panel allows for the manufacturing of a piece of furniture inheriting conductive elements which do not require any cables to power electric components. The most common example is an LED to provide interior lighting in furniture. A small model of an open cube was manufactured using the conductive MDF panel, a 5-layerd boards with 2 conductive layers (Figure 9a). The resulting object is a free-standing cube (Figure 9b) which can be combined with even more cubes (Figure 9c). Theoretically a light can be placed wherever the user wishes.

For the installation of the LED inside the furniture, a special LED fixation pin prototype was developed as previously shown by Tschannen et al. and Thoemen et al. [10,11]. The pin consists of two conductive metal parts which are connected to the anode and cathode of the LED and are separated by an electrical insulator. This pin can be pressed into a hole of the conductive MDF panel and positioned with very high flexibility. The power supply is connected by two wires to the conductive MDF panel by drilling a small hole for each cable into the cross-section of each conductive layer. For the small furniture in Figure 9b,c, the battery and the cables are mounted in a small pocket which has been machined into the bottom of the cube.

## 4. Conclusions

A homogenous mixture of carbon fibre in a wood-based panel is possible using the wet mixing process. The results of wet mixed wood fibre with carbon fibre are advantageous over dry mixed fibres in terms of the electrical properties. Additionally, the optical and microscopic assessment confirms the better mixing of wet mixed components. Comparing two different resin types also showed that much better results in terms of electrical properties are possible when using UF resin compared to pMDI. The carbon fibre content did not significantly influence the mechanical properties of wet and dry mixed panels but most definitely affected the electrical properties. The improvement of internal bond can be considered in a later study by improving the bonding between CF and resin. For this study, our emphasis was on the electrical properties, and hence UF resin was used instead of pMDI as the first assessment showed superior performance when assessing the electronic properties.

Using the described method allows for the manufacturing of 5-layerd conductive panels where the conductive layers are only 1.5 mm in thickness and thus allow for a very low carbon fibre content in the final board. The effective content for a perfectly working conductive MDF panel is less than 1 wt% of carbon fibres in relation to the dry wood mass of the MDF panel. One method to reduce the overall content of CF is to arrange them in a 5-layered structure with a very thin conductive layer. Furthermore, it was shown in this study that using the correct technique for mixing allows for a very homogeneous distribution of the CF and therefore reducing the overall quantity of CF needed to achieve a certain resistivity. The combination of mixing technique and arrangement of layers allow for reducing the quantity of CF to a minimum, allowing for a competitive advantage. Carbon fibres are a significant cost driver and reducing them to the absolute minimum is essential also in terms of sustainability. Ideally, carbon fibres from a renewable source could be used for this application since mechanical reinforcement is not the main interest.

## 5. Patents

The Patent WO2020244953A1—Structural element with electrically conductive properties and method for the use thereof resulted from the research described in this paper.

## Figures and Tables

**Figure 1 polymers-15-00912-f001:**
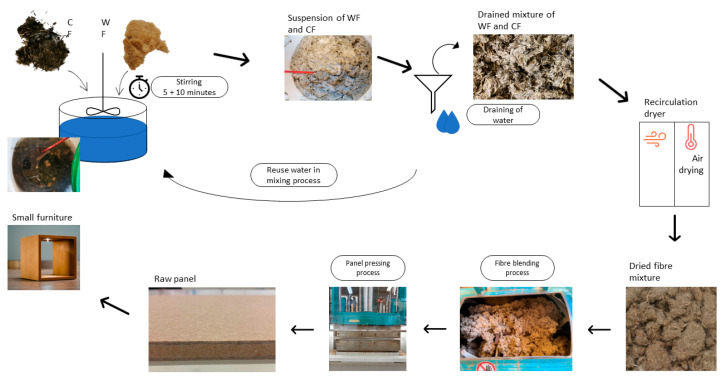
Schematic of wet mixing process of wood and carbon fibres for conductive MDF panel manufacturing.

**Figure 2 polymers-15-00912-f002:**
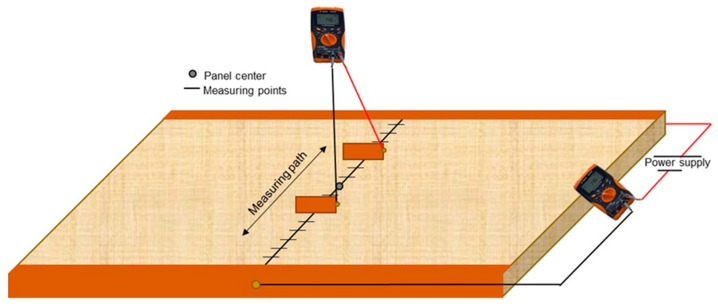
Schematic of resistance measurement process using the four-probe method with two multimeters and a power supply.

**Figure 3 polymers-15-00912-f003:**
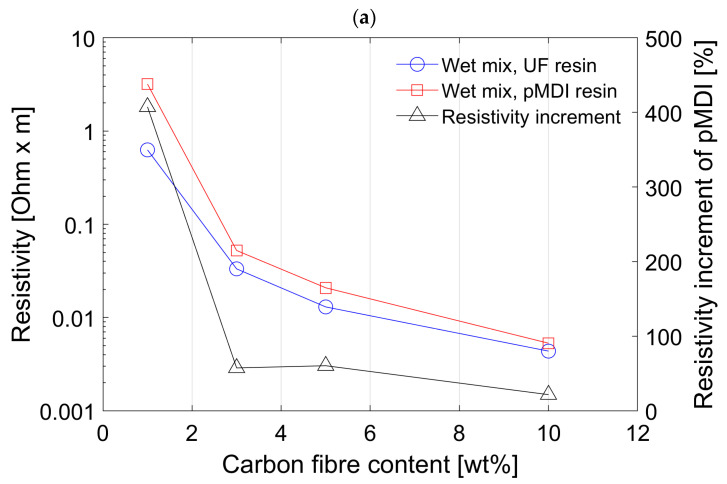
Mean resistivity of conductive MDF panel with different carbon fibre content and different resin types. (**a**) Comparing UF and pMDI resin. (**b**) Mixing process—comparing wet mixing with dry mixing process.

**Figure 4 polymers-15-00912-f004:**
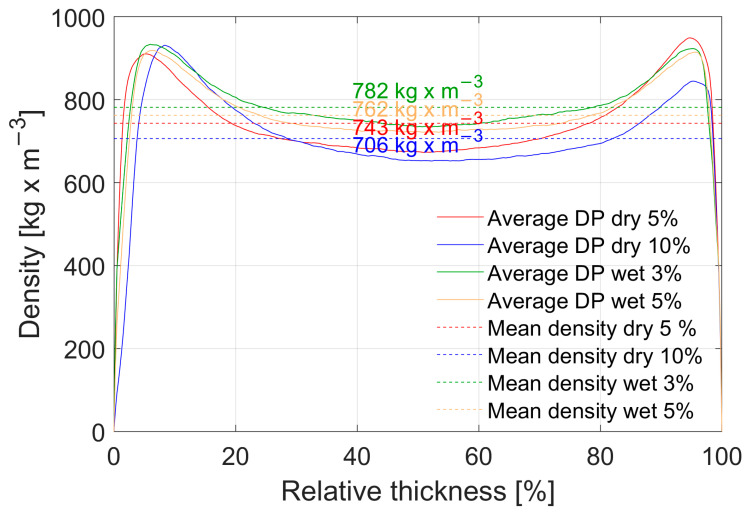
Density profiles of conductive MDF panels. The profile curve for each type of panel is showing an average density profile obtained from 9 measurements per type of panel. The dashed line indicates the mean density with its corresponding numerical value colour coded to each colour representing a type of panel.

**Figure 5 polymers-15-00912-f005:**
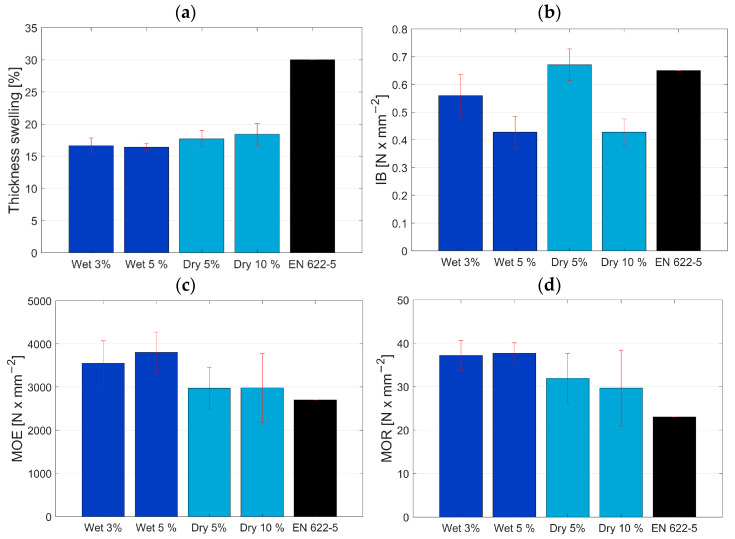
Results from mechanical and physical properties testing. Figures showing average, including standard deviation. (**a**) Thickness swelling—24 h, (**b**) internal bond, (**c**) modulus of elasticity, and (**d**) modulus of rupture. The black bar in the figure is representing the standard value for each property, respectively.

**Figure 6 polymers-15-00912-f006:**
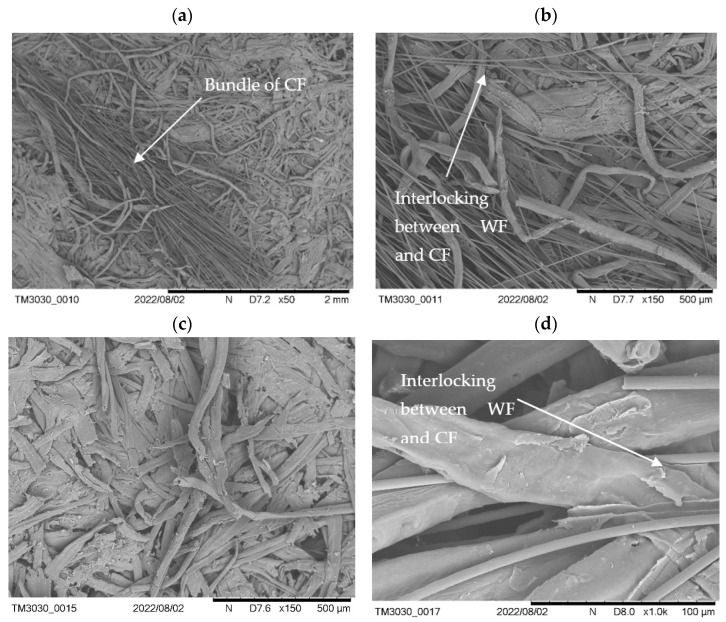
Scanning electron microscopy (SEM) pictures of dry mixed conductive MDF panels. Pictures taken from samples with 5 wt% CF after IB testing. (**a**) Agglomerate of CF. (**b**) Agglomerate of carbon fibres—interlocking CF with WF. (**c**) No visible CF. (**d**) Interlocking of CF with WF.

**Figure 7 polymers-15-00912-f007:**
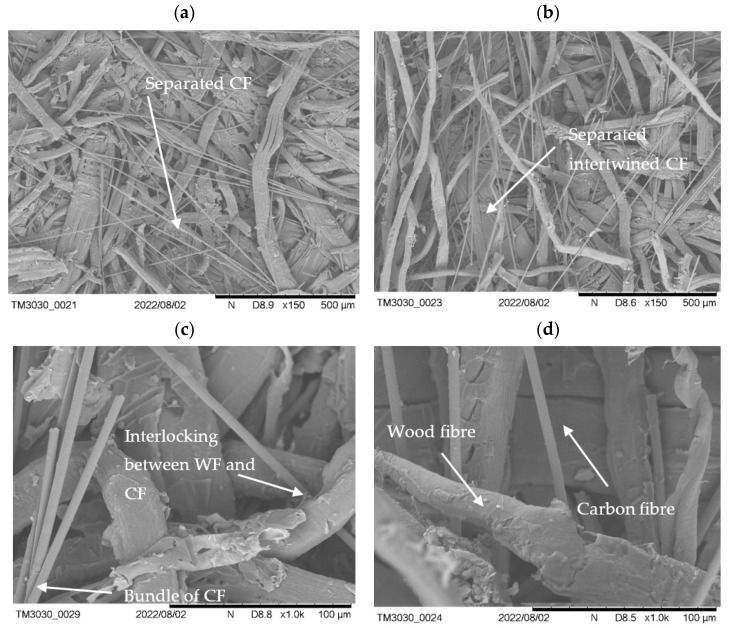
Scanning electron microscopy pictures (SEM) of wet mixed conductive MDF panels. Pictures taken from samples with 5 wt% CF after IB testing. (**a**) Separated CF with a homogeneous distribution. (**b**) Separated CF which are intertwined with WF. (**c**) Magnified view of CF and WF interlocking—smaller bundles of CF are still present. (**d**) Close-up of WF and CF.

**Figure 8 polymers-15-00912-f008:**
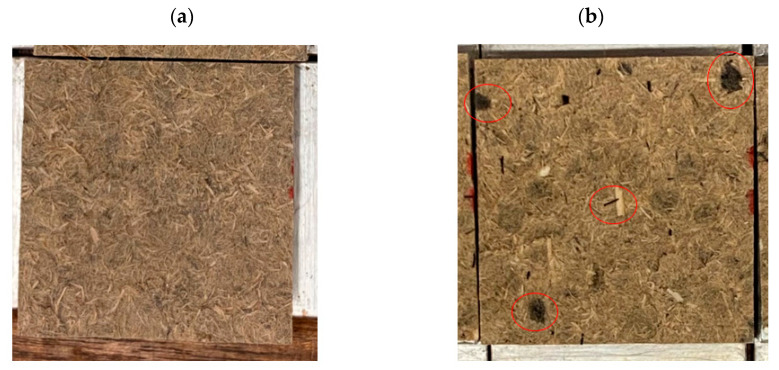
Breaking surface of IB samples after testing. Samples of (**a**) wet mixed panels and (**b**) dry mixed panels with red circles highlighting the agglomerates of CF. In both cases, the pictures have been taken from samples containing 5 wt% of CF.

**Figure 9 polymers-15-00912-f009:**
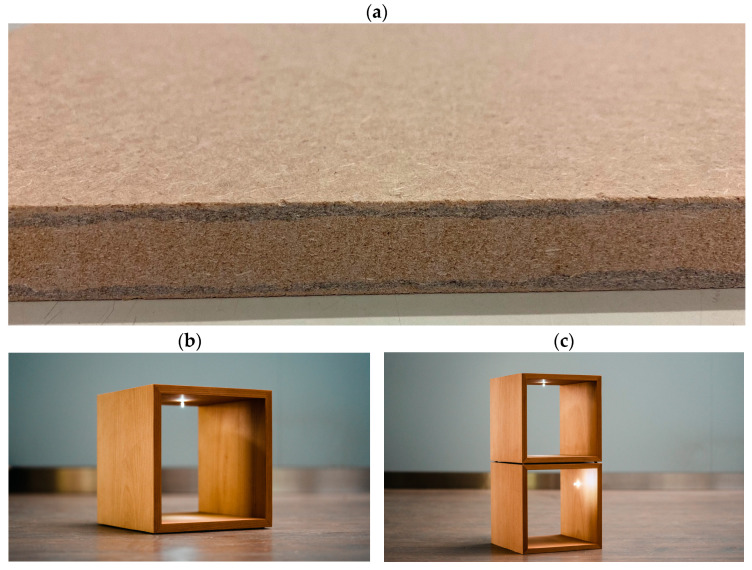
Furniture made from conductive MDF panels. Picture (**a**) showing the 5-layerd conductive MDF panel with conductive layers (dark in color) close to the surface, (**b**) a furniture cube made from conductive MDF panel and (**c**) individual cubes stackable with high flexibility with interior lighting made from conductive MDF panel.

**Table 1 polymers-15-00912-t001:** Panels variables investigated in this study.

Sample Code	Mixing Process	Resin Type	CF Content (%)
**A**	Wet mixing	UF	1
**B**			3
**C**			5
**D**			10
**E**		pMDI	1
**F**			3
**G**			5
**H**			10
**I**	Dry mixing	UF	5
**J**			10
**K (for furniture)**	Wet mixing	UF	5

**Table 2 polymers-15-00912-t002:** Conductivity of MDF panels at different carbon fibre content, resin type, and mixing process.

	Conductivity (S/m)			
Carbon Fibre Content (wt%)	UF-Board (wet mix)	pMDI-Board (Wet Mix)	UF-Board (Dry Mix)	Ratio (UF-Board Wet Mix/pMDI-Board Wet Mix/UF-Board Dry Mix)
**1**	1.59	0.31	-	1/5.1/-
**3**	30.06	19.10	-	1/1.6/-
**5**	76.93	47.97	24.85	1/1.6/3.1
**10**	229.20	188.50	109.53	1/1.2/2.1

## Data Availability

The data presented in this study is available upon request from the corresponding author.

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
