# Peer review of "Development of an Electrically Conductive MDF Panel—Evaluation of Carbon Content and Resin Type"

_polymers, 2023, doi:10.3390/polym15040912_

Round 1
Reviewer 1 Report
In this paper, highly conductive medium density fibreboard panels (MDF) were fabricated with wood and carbon fibres via two mixing processes (wet and dry), which can be applied in furniture and construction materials. The effect of resin type and carbon fiber content on the mechanical, physical and electrical properties of MDF were studied. The experiment was conducted professionally, and the discussion was adequate. Overall, this paper could be published in Polymers journal after all the comments are well addressed.
1. Section 3.1 of the manuscript elaborates that the CF applied in the experiment consists of a mixture of many different fibers and the conductivity is not known. It is suggested that the authors add this data.
2. It is suggested that the authors elaborate in the manuscript on how the values of solid content were determined for the two mixing processes (wet and dry).
3. The digital photo in Figure 8 does not show the difference between the fracture surfaces of the wet and dry blended samples. It is suggested to add a partial enlargement of the fracture pattern to better visualize the contrast.
4. In the last part of the manuscript, it is suggested that the authors describe how the MDF is connected to the power supply and LEDs?
Author Response
Dear Reviewer, thank you very much for your valuable comments and suggestions. Please find my detailed answers in the attached file.

Reviewer 2 Report
Dear Authors,
I found your manuscript as well prepared and of high potential to readers.
A few remarks mentioned below will help to make your manuscript better:
- line 120 - when you mixed the CF and WF in water, what was the fibres-to-water ratio?
- Figure 1 - In my opinion, you should add the additional box (step) for fibers blending with glue (probably between "dried fiber mixture" and "panel pressing process")
- line 153 - what was the maximum unit pressure during pressing the mats to get panels?
- line 260 (and other plots) - I suggest moving the information about the number of samples used to the Methodology part
Best regards!
Author Response

(The authors gave the same response as above.)

Round 2
Reviewer 1 Report
Accept in present form